# A Novel Combined Method for Measuring the Three-Dimensional Rotational Angle of a Spherical Joint

**DOI:** 10.3390/s24010090

**Published:** 2023-12-23

**Authors:** Qianyun Yang, Kai Ouyang, Long Yang, Rao Fu, Penghao Hu

**Affiliations:** Anhui Province Key Laboratory of Measuring Theory and Precision Instrument, School of Instrument Science and Opto-Electronic Engineering, Hefei University of Technology, Hefei 230009, China; 2021170006@mail.hfut.edu.cn (Q.Y.); 2020010001@mail.hfut.edu.cn (K.O.); yangl@mail.hfut.edu.cn (L.Y.); 2023110008@mail.hfut.edu.cn (R.F.)

**Keywords:** spherical joint, 3D rotation angle, RBF neural network

## Abstract

To improve the measurement accuracy of the three-dimensional rotation angle of a spherical joint, a novel approach is proposed in this study, which combines magnetic detection by a Hall sensor and surface feature identification by an eddy current sensor. Firstly, a permanent magnet is embedded in the ball head of a spherical joint, and Hall sensors are set and distributed in the ball socket to measure the variation in the magnetic flux density when the spherical joint rotates, which are related to the 3D rotation angle. In order to further improve the measurement accuracy and robustness, we also set grooves on the ball head and use eddy current sensors to synchronously identify the rotation angle of the ball head. After the combination of two signals is performed, a measurement model is established using the RBF neural network by training, and the real-time measurement of the 3D rotation angle of the spherical joint is realized. The feasibility and superiority of this method are validated through experiments. The experimental results indicate that the measurement accuracy is substantially promoted compared to the preliminary measurement scheme based on spherical coding; the average measurement error of the single axis is reduced by 9′9″. The root mean square errors for the measurements of the 3D rotation angles in this proposed method are as follows: pitch angle α has an error of 1′8″, yaw angle β has an error of 2′15″, and roll angle γ has an error of 29′6″.

## 1. Introduction

The spherical joint is a mechanism that can present three rotary degrees of freedom [1,2,3]. It is compact in structure and flexible in motion, and is widely used in parallel mechanisms, machine tools, measuring instruments, medical devices, optical devices, and various other equipment [4,5,6]. As a purely passive component, a spherical joint cannot determine its own rotation direction and rotation angle value [7,8]. If an embedded precise measurement method for its rotation angle can be obtained, the spherical joint becomes an intelligent device with broader application prospects, which is beneficial for improving the motion accuracy and facilitating the control of equipment for which spherical joints are used [9,10].

In recent decades, the identification of spherical rotation direction and angle measurement techniques have been researched and developed [11,12]. Several measurement methods based on different principles have emerged, which mainly include optical [13,14], magnetic [15], and inertial fields [16,17].

For example, Min Li utilized embedded sensors to simultaneously measure the magnetic flux density and back electromotive force of a spherical motor [18]. The data from these sensors were input into a sensor fusion system based on Kalman filtering to estimate the three degrees of freedom of angular displacement and the angle in real time. The idea for the implementation of this system is to use embedded non-contact sensors to measure the magnetic flux density of the spherical motor and the back electromotive force generated by the stator. Then, these two quantities are taken as two sets of decoupled inputs for the sensor fusion system based on the Kalman filter. This system includes an artificial neural network for estimating the rotor position and a mathematical model for calculating the angular velocity. Experimental results show that, within a certain measurement range, the average measurement error of the system in a single axis is 0.08°.

Jae-Hyeok Kim et al. proposed using a precision mechanical sensor called the attitude and heading reference system (AHRS), comprising a gyroscope, accelerometer, and magnetometer, to measure the tilt angle of a spherical motor [19]. The AHRS sensor is arranged on the ball joint. When the motor tilts, the sensor outputs a corresponding signal. By processing the signal, the tilt angle can be obtained. The measurement accuracy values of the two axes are 0.27° and 0.83°, but this method cannot measure the self-rotation angle of the motor, and the accuracy still needs to be improved.

Wang Q et al. also proposed a sensorless rotor attitude detection method based on the mutual inductance voltage of a stator coil [20]. In the online detection process of the rotor position, the three-dimensional angle of the rotor was inversely calculated based on the real-time collected mutual inductance voltage information, using an intelligent optimization algorithm, combined with the distribution law of the mutual inductance voltage and the constraints of the rotor structure. This detection method has a good online detection effect, with a standard deviation of the group within 1.8°; but, the accuracy is not high.

Yang S et al. proposed a two-degree-of-freedom angle displacement measurement method using a spherical capacitive sensor to measure a spherical pair [21]. The capacitance sensor proposed in this method had a four-quadrant differential electrode configuration. Compared with other angle measurement detection methods, it has an integrated structure, occupies a small space, and is convenient to integrate into the sphere. However, this method was unable to calculate measurements around the rotation axis.

Under the support of the National Natural Science Foundation of China, our team proposes a measurement scheme for three-dimensional rotational angles in the spherical joint space based on eddy current sensors and pseudo-random coding [22]. As shown in Figure 1a, a sensor array consisting of multiple sensors is used to identify spherical encoding. Two-dimensional spherical grooves are employed. The groove width is generated using pseudo-random encoding, and the groove depth is generated using an arithmetic sequence, ensuring the uniqueness of the spherical head’s three-dimensional encoding. An artificial neural network is employed to establish a measurement model between the output voltage of the eddy current sensors and the spatial three-dimensional (3D) rotational angles. Consequently, the measurement of 3D rotational angles in the spherical joint space is realized. In this scheme, the pitch angle α and the twist angle β are within the range of −10° to 10°, with root mean square errors of 22′32″ and 25′58″, respectively. The rotational angle, γ, of the spherical joint along the axis of the spherical joint rod is within the range of 0° to 120°, with a root mean square error of 30′17″. Figure 1b presents an alternative approach proposed by our team for measuring the two-dimensional rotational angle of a ball joint [23]. In this approach, a cylindrical permanent magnet is embedded at the bottom of the ball head, and a Hall sensor installed in the ball socket is used to measure the rotational angle of the ball head in any direction in space. Finally, the measured values are decomposed into rotational angle components α and β around the *X*- and *Y*-axes, respectively. This approach has two modeling methods: one is to establish an equivalent magnetic charge model, and the other is to establish a neural network model. The experimental results demonstrate that the neural network model has higher accuracy, a simpler structure, faster data processing speed, and the highest single-axis angle measurement accuracy can reach 4′. However, this method has a lower accuracy for measuring the rotational angle, γ, of the spherical joint along the axis of the spherical joint rod; thus, it can be considered unsuitable for measuring the γ angle.

The accuracy level of the measurement scheme for the three-dimensional rotation angle of the spherical joint space based on eddy current sensors and pseudo-random codes is not able to meet the needs of the precision engineering field. Therefore, in order to solve the problem that the measurement system of the spherical hinge based on the magnetic effect method cannot effectively measure the rotational angle, γ, of the spherical joint along the axis of the spherical joint rod, this article combines the advantages of the magnetic effect and spherical coding methods based on previous research and constructs a new combined measurement scheme.

## 2. Measurement Plan Design

This scheme embeds a permanent magnet into the bottom of the ball head and uses three Hall sensors to detect the magnetism of the ball head. When the spherical joint rotates, the Hall sensors perceive the 3D rotation angle of the ball hinge through the change in the magnetic field. However, this scheme has a low measurement accuracy for the roll angle, γ. Therefore, a one-dimensional groove was machined on the surface of the metal ball head to improve the measurement accuracy of the self-rotation angle, γ, using the distance measuring principle of eddy current sensors, and also to enhance the measurement accuracy of the pitch angle, α, and yaw angle, β. The overall design is shown in Figure 2. This measurement scheme limits the three translational degrees of freedom along the *X*-, *Y*-, and *Z*-axes of the ball hinge, and does not limit the three rotational degrees of freedom around the *X*-, *Y*-, and *Z*-axes.

### 2.1. Sensor Placement Location

First, three Hall sensors were horizontally placed at the bottom of the ball socket. Then, the sensor positioning fixture inside the ball socket was used to fix the three Hall sensors in the same plane. Two different sensor placement schemes are used here for comparison. Scheme 1: as shown in Figure 3a—sensor S1 is located on the *X*-axis and senses the magnetic field in the *X*-axis direction, sensor S2 is located on the *Y*-axis and senses the magnetic field in the *Y*-axis direction, and sensor S3 is located on the axis that is 135°counterclockwise from the *Y*-axis, sensing the magnetic field in the axial direction. Scheme 2: as shown in Figure 3b—sensors S1 and S2 are rotated by 90°, respectively. S1 measures the magnetic field component in the *Y*-axis direction and S2 measures the magnetic field component in the *X*-axis direction. The position of S3 remains unchanged. The coordinates in Figure 3 represent the position of the sensor in a plane coordinate system.

Using COMSOL and MATLAB, simulations were conducted for two different schemes. During the finite element simulation process in COMSOL, the γ angle remained constant, while the α and β angles varied within a range of ±20°. An RBF (radial basis function) neural network model was constructed in MATLAB, and the simulated α and β angles were inputted to fit the error results, as shown in Figure 4. It can be seen that scheme 2 has higher measurement accuracy results for the α and β angles, so scheme 2 is adapted to place the Hall sensor. The design of the ball socket is shown in Figure 5. Based on the spherical coding scheme described above, the eddy current sensor was installed at any four asymmetric positions on the surface of the spherical socket, which could improve the measurement accuracy of the three-dimensional rotation angle of the spherical joint to a certain extent [22].

### 2.2. Spherical Groove Design

Using the AD/DC module of the COMSOL physics field simulation tool, a three-dimensional finite element model was established, as shown in Figure 6b. The colored graphics on the sensor represent the magnitude of the magnetic induction, the direction of the arrows represents the direction of the current, and the color of the arrows represents the magnitude of the current. From the simulation results, it can also be seen that the eddy current field is mainly distributed around the sensor, and the apparent range of the eddy current field does not exceed the size of the sensor. This also indicates that the width of the grooves and protrusions should not exceed the size of the sensor; otherwise, there is a “blind spot” in the measurement.

At the same time, the simulation also shows that the direction of the eddy current is opposite to the direction of the sensor current. From Figure 6a, it can be seen that the magnetic field generated by the magnet is distributed around the magnet, and the farther away from the magnet, the weaker the magnetic field. The eddy current sensor can only sense the magnetic field around the sensor, and the influence on the eddy current sensor is minor when the magnet is far away from the eddy current sensor. Moreover, since the magnetic field around the magnet is a uniform demagnetized field, this fixed influence is beneficial to the measurement of the angle when we need the eddy current sensor to generate specific data to identify the angle.

In addition, by simulating and analyzing the grooves with different parameters on a metal block, the parameter settings of the grooves on the surface of the spherical head were determined. Based on the research results for the output characteristics of the eddy current sensors, it is known that the inductance of the sensor will undergo significant changes when scanning grooves with different parameters, especially when the sensor is located at the center of the groove, where the difference value is the maximum [24]. For grooves with the same width, the larger the groove depth, the greater the change in inductance; for grooves with the same depth, the larger the groove width, the greater the change in inductance. Additionally, the sensor is more sensitive to the groove depth than groove width.

Therefore, in this study, only the parameter setting of the groove depth was modified. Finally, through a large number of combination schemes, the optimal solution was determined as follows: the spherical head was divided into 24 groups, with each group consisting of a 15° interval, and the groove width occupied 7°l (where l was the arc length when the central angle of the equatorial plane of the sphere was 1°). Among these groups, 12 groups had groove depths starting from 0.1 mm and increased by 0.1 mm each time, while the other 12 groups ad groove depths starting from 1.25 mm and decreased by 0.1 mm each time. Based on the design scheme described above, the physical structures of the ball head and ball socket are shown in Figure 7. The ball head was made of aluminum alloy and the ball socket was manufactured using 3D printing with nylon material, which does not affect the sensor measurements.

## 3. RBF Neural Network

Based on the team’s previous experience of using neural network modeling [22], establishing a measurement model for the rotational angle of the spherical joint space based on artificial neural networks can simplify the algorithm model, eliminate the complex and lengthy model derivation process, and the high robustness of the neural network can also compensate for the defects in prototype structural parameters, installation errors, and gap errors during ball head movements. Among them, the RBF neural network can approximate any nonlinear function with arbitrary precision and has a good generalization ability. When the network parameters are determined, the output of the network is the linear weighted sum of the hidden layer node outputs, so various linear optimization algorithms can be used to solve the network weights, speed up the learning speed, and avoid local minimum value problems [25,26,27].

The RBF neural network structure, as shown in Figure 8, is composed of an input layer, hidden layer, and output layer. The transformation from the input layer space to the hidden layer space is non-linear, while the transformation from the hidden layer space to the output layer space is linear.

The network input is the output of the Hall sensor and eddy current sensor, denoted as X = [*x*_1_, *x*_2_, …, *x*_7_] ^T^. The network output is the predicted values of three rotational angles, denoted as Y = [ *y*_1_, *y*_2_, *y*_3_] ^T^. The expression of the output layer of the RBF network is:(1)$$y\left(x_{i}\right)=\sum_{i=1}^{l}\omega_{i}h_{i}\left(x\right)\;$$
(2)$$h\left(x\right)=\exp\left(-\frac{{\left|\left|x-c_{i}\right|\right|}^{2}}{\sigma_{i}^{2}}\right),i=1,2,\ldots ,l$$
where $\omega_{i}$ is the ith output weight vector and l is the number of nodes in the hidden layer;$\;h_{i}\left(x\right)$ is the activation function and the Gaussian function is the most commonly used radial basis function; and $c_{i}$ represents the center parameter of the kernel function for the ith hidden layer neuron, while $\sigma_{i}$ is the expansion constant for the ith hidden node.

Due to the fact that the center point selection of the RBF algorithm uses the K-means algorithm, the training process adjusts the weights of the network using either gradient descent or least squares method, which often leads to overfitting and reduces the model’s generalization ability. While the RBF neural network optimizes the centers of the hidden layer ($c_{i}$), expansion constants ($\sigma_{i}$), and output weights ($\omega_{i}$) as particles in the particle swarm algorithm, this approach effectively avoids overfitting and other problems that may arise during model training [28,29,30]. Therefore, this paper used the PSO algorithm to optimize the RBF neural network, improved its robustness and generalization ability, and enhanced the accuracy of the measurement system.

The optimization process of the PSO (particle swarm optimization) algorithm for RBF neural networks can be roughly divided into the following steps. First, the particle swarm is generated by determining the structure of the RBF neural network. The particles are then mapped to the RBF neural network, establishing the RBF neural network model. Then, the expansion constant ($\sigma_{i}$) is calculated using Formula (3). Based on the expansion constant ($\sigma_{i}$) values, the global best particle and individual best particle are updated, and then it is determined whether the threshold targets are met. If the requirements are not met, the velocity and position of the particles are updated and the particle swarm is regenerated. If the requirements are met, the optimal particle is outputted and the particles are mapped to the RBF neural network for testing.

The calculation formula of the extended constant is:(3)$$\sigma =\frac{d_{max}}{\sqrt{2n}}$$
where $d_{max}$ represents the maximum distance between the selected centers. n represents the number of samples.

The optimization goal of the PSO algorithm is to minimize the error function value between the actual output and the expected output of the RBF neural network [31]. The fitness function is set as the objective:(4)$$F=\sum_{i=1}^{n}\sum_{j=1}^{e}\left(d_{ij}-o_{ij}\right)\;$$
where n represents the number of samples, e represents the number of outputs of the neural network, $d_{ij}$ represents the j expected output of the i sample of the RBF neural network, and $o_{ij}$ represents the j actual output of the i sample of the RBF neural network.

The RBF and PSO optimized RBF neural network models were established. The data of sensors S1, S2, and S3 obtained from the simulation of scheme 2 in Section 2.1 were used as inputs, and the corresponding alpha and beta angles were used as outputs. The RBF neural network and the improved PSO optimized RBF neural network were tested. In order to reduce the impact of randomness on the network performance, 50 simulation tests were conducted.

The test results are shown in Figure 9. From the graph, it can be observed that, for each simulation, the fitting effect of the PSO optimized RBF neural network model is superior to the unoptimized RBF neural network. The PSO optimized RBF neural network model is capable of controlling the root mean square errors of the α and β angles to around 0.09°, whereas the unoptimized RBF neural network model yields an RMSE of the α and β angles of around 0.16°. The reason why the unoptimized results in the graph appear as a straight line is because this study utilizes MATLAB’s newrb function for the RBF neural network. Under the condition of a consistent training set and unchanged sample order, the results of multiple training using this function are consistent and not affected by randomness.

Through the validation of the γ angle with the COMSOL simulation data in Section 2.2, it can be seen in Figure 10 that the error distribution of the optimized RBF neural network is similar to that of the unoptimized RBF neural network, with a slight improvement in the accuracy. Through the data analysis, the improved PSO algorithm optimized the maximum error of the γ angle tested by the RBF neural network model to be 1.68°, with a root mean square error of 0.52°, while the unoptimized RBF neural network model tested a maximum error of 1.73° for the γ angle, with a root mean square error of 0.59°.

## 4. Experiment

### 4.1. Experimental Equipment

The three-dimensional diagram of the experimental setup, as shown in Figure 11, includes three calibrated rotary stages for rotation around the *X*-, *Y*-, and *Z*-axes, namely, the RPI (rotary precision instrument), LS (Lian Sheng), and PI (Physik Instrumente) stages, with respective accuracies of ±1″, ±4″, and ±2″. The ball head was composed of an aluminum alloy material with a diameter of 50 mm. The permanent magnet used was a cylindrical sintered neodymium iron boron magnet, with a residual magnetic parameter of 1.2 mT and magnetization direction along the axis, with a diameter of 15 mm and height of 5 mm. The Hall sensor had a measurement range of 0~3 T and a resolution of 10–5 mT. The eddy current sensor had a measurement range of 1.5 mm and a resolution of 0.15 um.

The experimental process was carried out according to the rotation sequence of Euler angle x–y–z, first rotating the RPI table (α angle), then rotating the LS table (β angle), and finally rotating the PI table (γ angle).

### 4.2. Experimental Data Analysis

A total of 21*21*181 sets of the data were collected in the experiment, with a measurement range of ″−10°≤″ α ≤ 10°, ″−10°≤″ β ≤ 10°, ″−90°≤″ γ ≤ 90° and a sampling interval of 1°. Due to the amount of data, inputting all the data into the network resulted in excessively long training times, so the data were partitioned. Based on γ, each 20° was taken as an independent region, with a total of 9 regions. Table 1 shows the test results for each region, ME represents the mean error, and RMSE represents the root mean square error. It can be seen that the results between regions are not significantly different. The average difference in the RMSE value of the three rotation angles is less than 1′.

By integrating the measurement results from nine regions, the RMSE values of the rotation angles α, β, and γ are 1′8″, 2′15″, and 29′6″, respectively. The test results of all regions are shown in Figure 12, where the color represents the magnitude of the error. Due to the large number of test points, only points with significant errors are shown in this figure. Figure 12a shows the test error of the α angle in all regional test sets. For the α angle, most of the errors are below 0.1°. The errors in each region are relatively balanced, and the points with large errors are concentrated near 0° for α, β, and γ. Figure 12b shows the test errors for the beta angle of the test sets in all regions. For the beta angle, most of the errors are below 0.4°. As with the alpha angle, the larger errors are concentrated near 0° for α, β, and γ. Figure 12c shows the test errors of the γ angle in all regional test sets. Most of the errors of the γ angle are below 1°, which is equivalent to α and β angles with large errors and uniform error distributions.

## 5. Analysis of Uncertainty

From the perspective of research progress and experimental experience, the main error factors affecting the measurement accuracy of the three-dimensional rotation of the spherical joint included the error of the permanent magnet structure parameters and magnetic field eccentricity, the eccentricity error of the ball head, the rotational error of the turntable, the error of repeated measurements, the stability error of the sensor, and the fitting calculation error of the neural network mode. Based on these sources of error, several types of uncertainty sources can currently be observed and verified through calculations.

### 5.1. Uncertainty Introduced by Sensor Measurement Repeatability

Under the same experimental conditions, measurements were repeated 10 times at the same spherical position (5°, 5°, 5°). Based on the results of the 10 repeated measurements, the uncertainty components introduced by the measurement repeatability error were evaluated, as shown in Table 2.

The standard uncertainty calculation formula for repetitive errors is as follows [32]:(5)$$u_{R}=\sqrt{\frac{\sum_{i=1}^{n}{\left(a_{i}-\bar{a}\right)}^{2}}{10\times 9}}$$
where $u_{R}$ is the uncertainty introduced by sensor measurement repeatability, $a_{i}$ is the respective angular value for the i measurement, and $\bar{\;a}$ is the average value of the angles. The calculation results are shown in Table 3.

### 5.2. Uncertainty Introduced by Drift

With the continuous changes in the experimental environment (temperature, vibration, etc.), the measurement system experienced a drift in the parameters of its components or mechanisms during the operation, which affected the accuracy of the measurement results. To assess the uncertainty introduced by drift to the system, the measurement system was kept stationary in a constant-temperature laboratory for a while, and the system’s output values were recorded in real time. The uncertainty was then evaluated by calculating the range difference of the drift data during this period.

Under the assumption that the drift error follows a uniform distribution, the formula for calculating the standard uncertainty is as follows [32]:(6)$$u_{D}=\frac{a}{\sqrt{3}}$$
where $u_{D}$ is the uncertainty introduced by drift and a is the maximum value of the error of the measured value. The probability that the measurement error falls within the interval (x−a,x+a) is 1. The calculation and results are shown in Table 4.

Due to the fact that the two uncertainties are caused by different errors, it can be considered that they are independent of each other. Therefore, the formula for the combined standard uncertainty is [32]:(7)$$u_{C}=\sqrt{\left(u_{R}^{2}+u_{D}^{2}\right)}$$

The synthetic degrees of freedom is [32]:(8)$$v_{C}=\frac{u_{C}^{4}}{\frac{u_{R}^{4}}{v_{R}}+\frac{u_{D}^{4}}{v_{D}}}$$

The combined uncertainty results are shown in Table 5.

The confidence probability was set to *p* = 0.95, we determined the inclusion factor, k, by checking the t-distribution table through the degrees of freedom, and calculated the spreading uncertainty. The results are:(α, β, γ) = (4.94932, 4.96586, 4.48796) ± (0.015191, 0.017381, 0.078356)(9)

## 6. Conclusions

This article presented a new method for the precise measurement of the 3D rotation angles of a spherical joint based on Hall and eddy current sensors. The spatial position matching of the permanent magnet and Hall sensor was optimized. The relationship between the output of the eddy current sensor and groove parameters was explored to determine the appropriate groove scheme for the ball head. A measurement model was established using the PSO-RBF neural network algorithm, and the feasibility of the method was verified through experiments. The three-dimensional rotation angles of the spherical joint were measured, with the root mean square errors of rotation angles α, β, and γ being 1′8″, 2′15″, and 29′6″, respectively, and the mean errors being 51″, 1′20″, and 22′57″. Compared with the spherical encoding three-dimensional rotation angle measurement scheme with root mean square errors of 22′32″, 25′58″, and 30′17″, the accuracy was significantly improved.

The innovation of this method lies in the combination of the eddy current and Hall sensors, which solves the problem that the two-dimensional rotation angle scheme of ball hinge based on a magnetic effect cannot effectively measure rotation angle γ on the rotation axis. This scheme enabled the measurement of the three-dimensional rotation angle and provided a new method for the measurement of the three-degree-of-freedom rotation angle. On this basis, the PSO-RBF neural network algorithm was used to improve the precision of the spherical hinge measurement system. This scheme presents benefits for industrial precision manufacturing applications because it can work in harsh environments, such as situations where grating measurements are difficult to perform. It has excellent environmental adaptability, high accuracy, and fast response results, among other advantages. When implementing this scheme, there are no limitations on the required dimensions. As long as it is within the processable range of the spherical head, this scheme can be successfully used, and it has a wide range of applications. Compared to previous sensor modeling methods, this scheme simplifies the establishment of measurement models and reduces the impact of data processing on measurement accuracy.

## Figures and Tables

**Figure 1 sensors-24-00090-f001:**
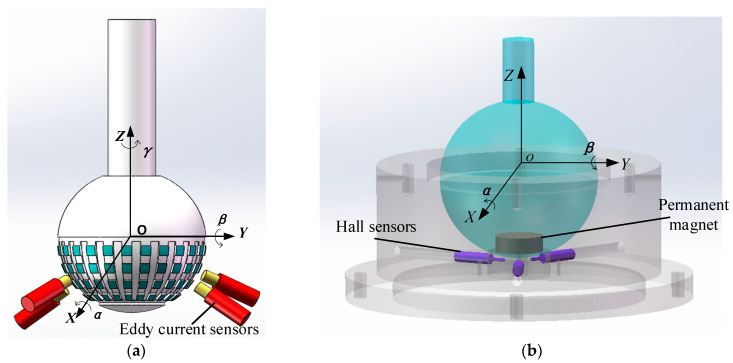
Two measurement schemes (**a**,**b**) for the rotation angle of a spherical joint.

**Figure 2 sensors-24-00090-f002:**
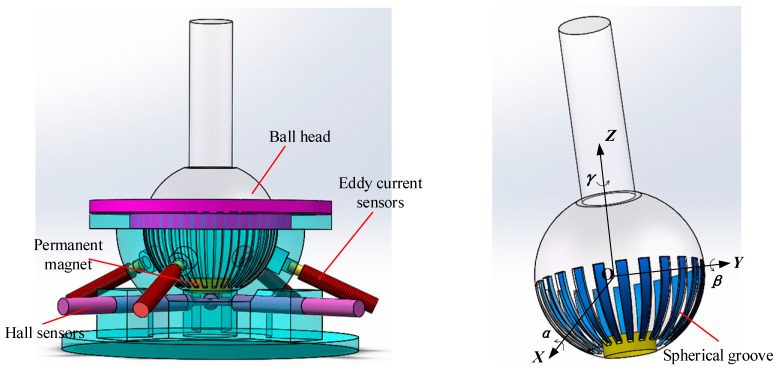
Schematic diagrams of measurement scheme.

**Figure 3 sensors-24-00090-f003:**
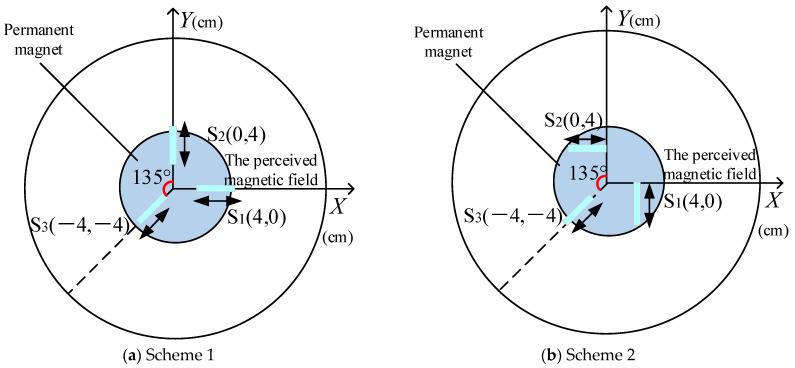
Hall sensor-placement angle design.

**Figure 4 sensors-24-00090-f004:**
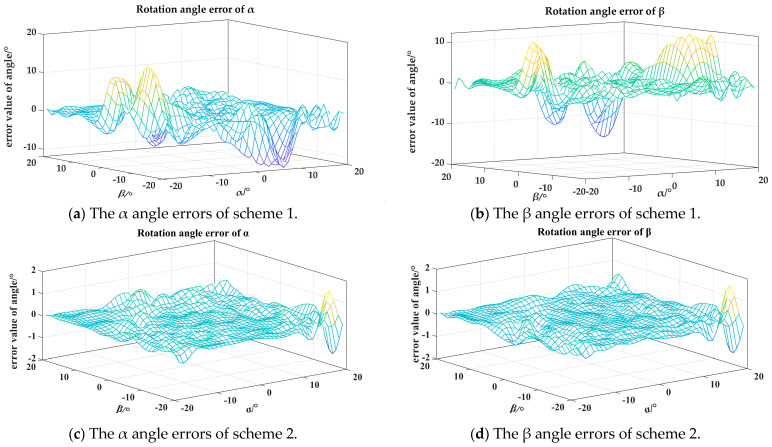
The α and β angle errors of the neural network fitting process.

**Figure 5 sensors-24-00090-f005:**
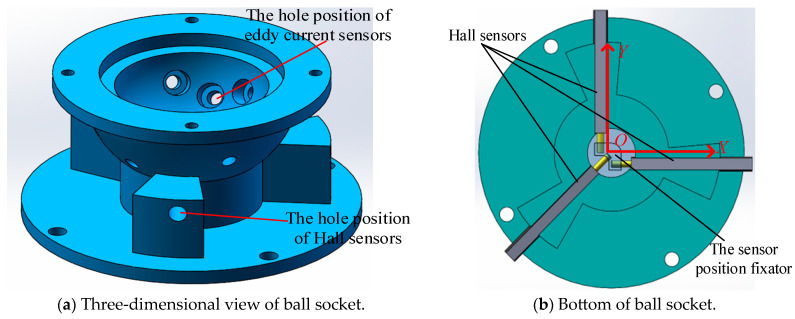
The design scheme of the ball socket.

**Figure 6 sensors-24-00090-f006:**
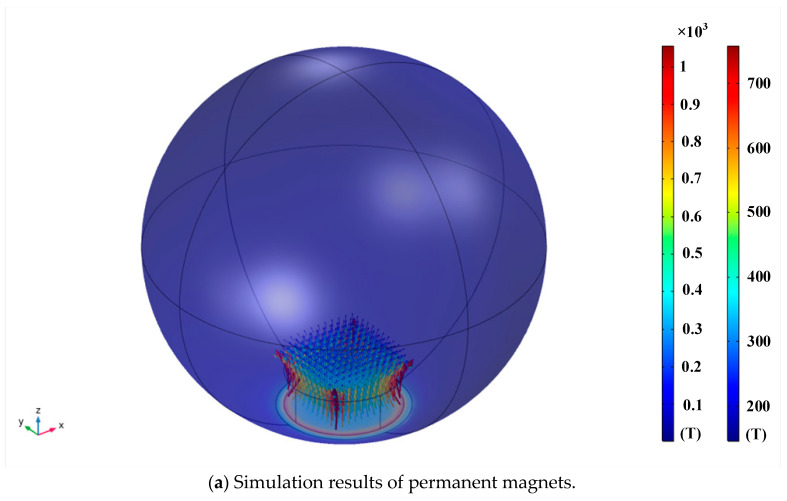
Simulation results based on COMSOL.

**Figure 7 sensors-24-00090-f007:**
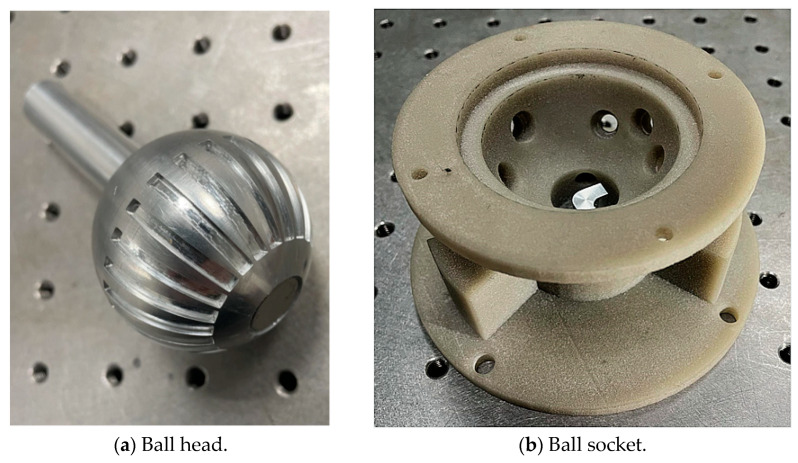
Physical diagrams of the ball head and socket.

**Figure 8 sensors-24-00090-f008:**
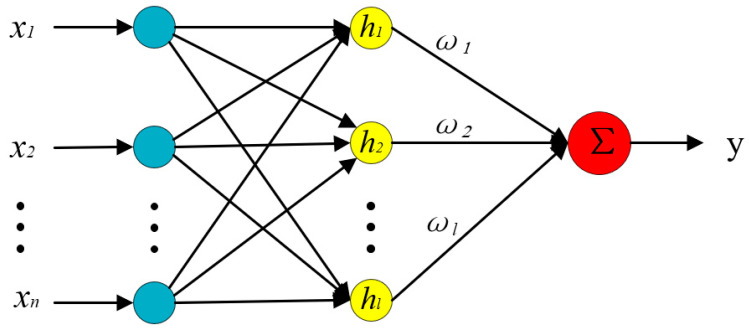
Structure of the RBF neural network.

**Figure 9 sensors-24-00090-f009:**
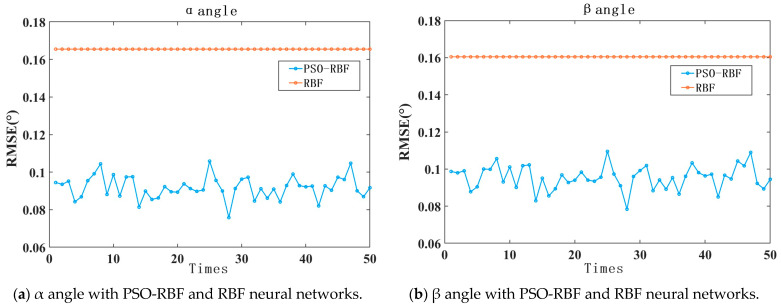
The comparison of test results for α and β angles with the PSO optimized RBF neural network and unoptimized RBF neural network.

**Figure 10 sensors-24-00090-f010:**
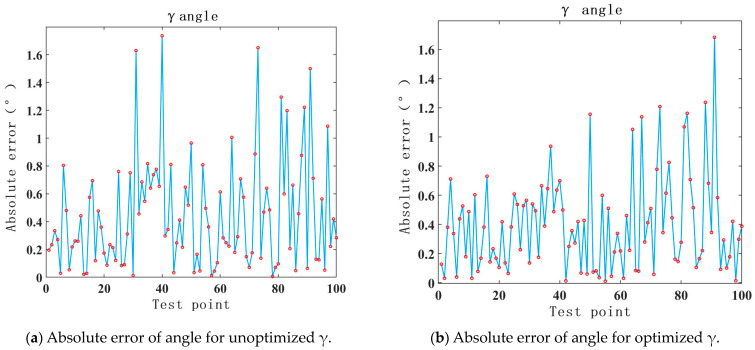
The comparison of test results for γ with the PSO optimized RBF neural network and unoptimized RBF neural network.

**Figure 11 sensors-24-00090-f011:**
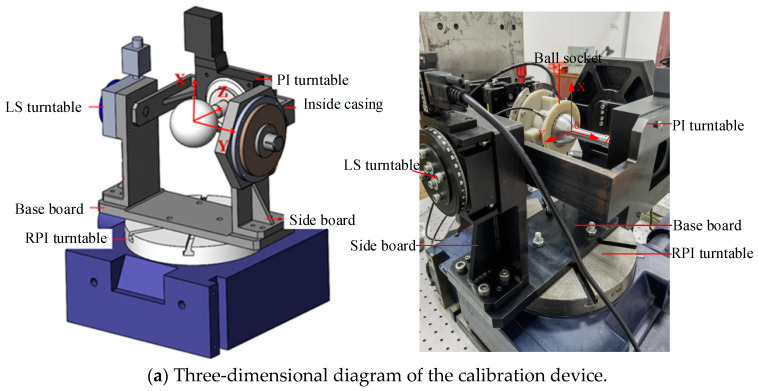
Experimental installation.

**Figure 12 sensors-24-00090-f012:**
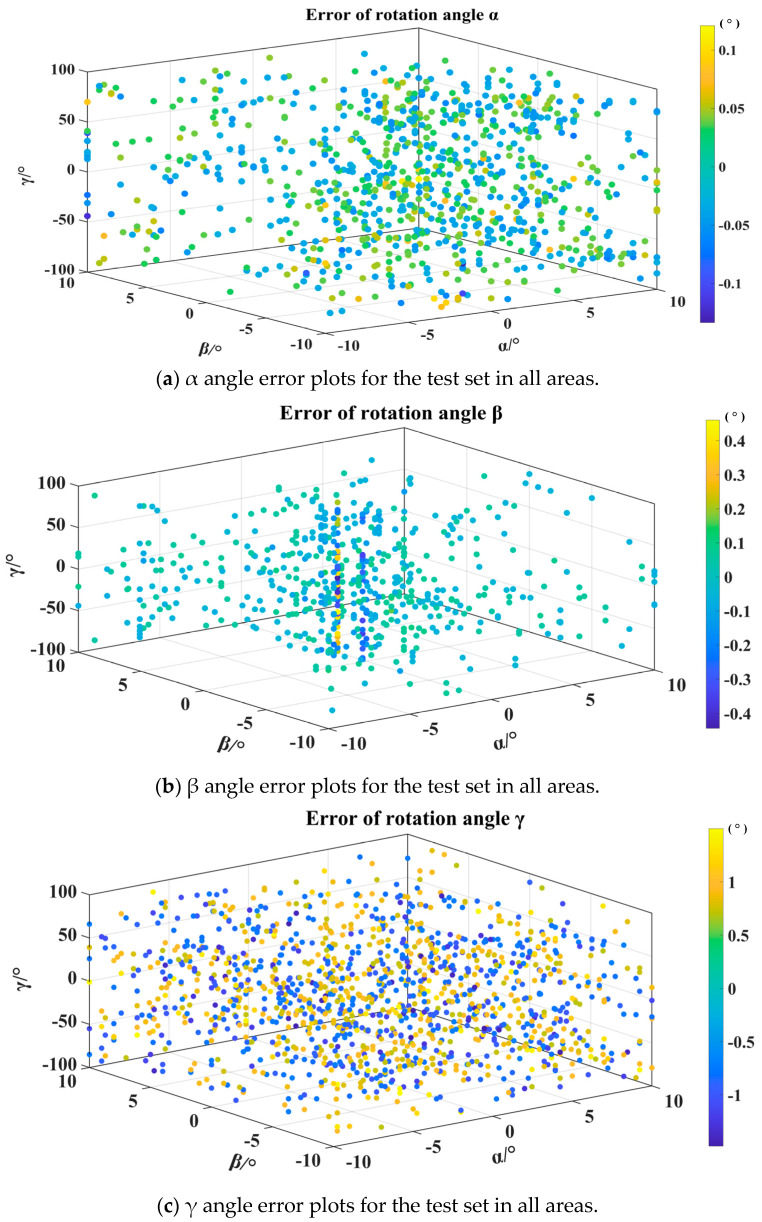
Three rotation angle error plots for the test set in all areas.

**Table 1 sensors-24-00090-t001:** Comparison of test results across all regions.

	*α*		*β*		*γ*	
	ME	RMSE	ME	RMSE	ME	RMSE
Q1	50.4″	1′22.8″	1′19.2″	2′9.6″	22′15.6″	28′44.4″
Q2	61.2″	1′19.2″	1′22.8″	2′20.4″	23′31.2″	29′38.4″
Q3	50.4″	1′8.4″	1′26.4″	2′34.8″	23′24″	29′49.2″
Q4	46.8″	1′4.8″	1′26.4″	2′31.2″	23′34.8″	29′52.8″
Q5	46.8″	1′1.2″	1′26.4″	2′24″	23′38.4″	30′7.2″
Q6	43.2″	57.6″	1′19.2″	2′20.4″	22′44.4″	28′37.2″
Q7	54″	1′12″	1′15.6″	1′48″	23′13.2″	29′24″
Q8	54″	1′8.4″	1′8.4″	1′40.8″	21′46.8″	27′32.4″
Q9	54″	1′12″	1′15.6″	1′58.8″	22′26.4″	28′8.4″

**Table 2 sensors-24-00090-t002:** Repeatability measurement data for (5°, 5°, 5°).

	1	2	3	4	5	6	7	8	9	10
α/°	4.9264	4.9364	4.9344	4.9264	4.9625	4.9731	4.9311	4.9645	4.9564	4.9820
β/°	4.9689	4.9785	4.9805	4.9689	4.9536	4.9566	4.9720	4.9431	4.9513	4.9852
γ/°	4.3970	4.5035	4.5538	4.3970	4.557	4.6025	4.4078	4.5124	4.6031	4.3395

**Table 3 sensors-24-00090-t003:** Standard uncertainty components introduced by repeatability.

Rotation Angle	$\bar{a}$/°	$u_{R}$/°	$v_{R}$
α	4.94932	0.006549	9
β	4.96586	0.004435	
γ	4.48736	0.030254	

**Table 4 sensors-24-00090-t004:** Standard uncertainty components introduced by drift.

Rotation Angle	a/°	$u_{D}$/°	$v_{D}$
α	0.0038	0.00219	5305
β	0.0133	0.00768	
γ	0.0394	0.02275	

**Table 5 sensors-24-00090-t005:** Combined uncertainty.

Rotation Angle	$u_{C}/{}^{\circ}$	$v_{C}$
α	0.006905	11
β	0.008868	141
γ	0.037853	22

## Data Availability

Data are contained within the article.

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
