# Peer review of "A Novel Combined Method for Measuring the Three-Dimensional Rotational Angle of a Spherical Joint"

_sensors, 2023, doi:10.3390/s24010090_

Round 1

Reviewer 1 Report

Comments and Suggestions for Authors

In this paper, a new method based on Hall sensor and eddy current sensor is proposed for the accurate measurement of three-dimensional Angle of spherical joint. The spatial position matching between permanent magnet and Hall sensor is optimized. The relationship between eddy current sensor output and groove parameters is discussed, and a suitable groove scheme for ball head is determined. The measurement model is established by using PSO-RBF neural network algorithm, and the feasibility of the method is verified by experiments. But I still have some comments and suggestions.

1.      1. Please check the pictures carefully. For example, the horizontal coordinates of the pictures (a) and (c) in Figure 4 are incomplete; The clarity in Figure 6 is not high enough.

2.      “In addition, by simulating and analyzing grooves with different parameters on a metal block, the parameter settings of the grooves on the surface of the spherical head were determined.” Please specify the specific parameter setting of the grooves on the surface of the ball head.

3.      The paragraph below Figure 6 says: “Based on the research results on the output characteristics of eddy current sensors, it is known that the inductance of the sensor will undergo significant changes when scanning grooves with different parameters, especially when the sensor is located at the center of the groove, the difference value is maximum. For grooves with the same width, the larger the groove depth, the greater the change in inductance; for grooves with the same depth, the larger the groove width, the greater the change in inductance. Additionally, the sensor is more sensitive to groove depth than groove width.” Ask the author to explain the research results on which this conclusion is based and provide simulation data to support your opinion.

4.      “The sensor is more sensitive to groove depth than groove width. Therefore, in this study, only the parameter setting of groove depth is modified.” I think this is not rigorous, affecting the results not only the depth of the slot, but also the coordination between the two, can not only consider the depth of the slot.

5.      This paper does not include a single MDPI paper in its references. Please include relevant MDPI papers as references.

6.      It is better if the language in the text is polished.

Reviewer 2 Report

Comments and Suggestions for Authors

The authors propose an original method to measure the configuration of the spherical joint using two sets of sensors. The presented material is illustrative and easy follow. On the other hand, there are several unclear terms and sentences to be revised. Some notations have a nonuniform style either. Furthermore, the experimental results can be misleading because of the incorrect measuring scheme. Please see the comments below for the details.

Major comments:

1.        Page 4, subsec. 2.1. It is unclear if angle γ affects the results shown in Fig. 4. Do the authors keep a constant value of this angle or vary it during their simulations?

2.        P. 8, Eqs. (1) and (2) and the paragraphs around them. The notations used in the text and equations are inconsistent. The style of variables changes from roman to italic and from lowercase to uppercase (X and x), some numbers should be typed as subscripts (x1 and x1), and some subscripts are missed (h and hi, σ and σi, and so on). The authors should carefully revise the notations and unify their style.

3.        P. 8, the RBF neural network:

3.1.       It is unclear how the authors train the network. Did the authors get the training set from the COMSOL model?

3.2.       What are the obtained network parameters (specified in Eqs. (1) and (2)) for the optimized and non-optimized models?

3.3.       The description of the PSO algorithm appears after the simulation results (p. 9). It would be more natural to place it in the text before these results. In addition, this description looks too short and poorly connected with the preceding part. I recommend the authors revise and extend the material devoted to the PSO algorithm and, if possible, use notations and provide equations related to Eqs. (1) and (2).

4.        P. 10–11, the experimental section. According to the experimental platform (Fig. 11), the Y axis (LS turntable) becomes rotated after the rotation of the RPI turntable around the X axis. This means the LS turntable controls the rotation and specifies angle β about the rotated Y axis. On the other hand, the preceding theoretical section showed angles α and β were measured about the fixed X and Y axes (Fig. 1). To implement the proposed methods, the authors should deal with this issue and recompute one rotation angles from the others. The paper, however, does not address this important point. As a result, all the obtained results can be misleading.

5.        P. 15–16, the reference list. The authors should carefully revise their reference list:

5.1.       The references have a nonuniform style, especially in the information about the authors. Some papers have both the first and the second names of the authors, while the others have only the second names. Furthermore, some papers incorrectly spell the authors’ names or have the wrong order of the authors (for example, [3], [10], [18], and possibly other.)

5.2.       Refs. [1] and [2], [6] and [9] doubles each other.

5.3.       Refs. [3] and [11] have no page numbering.

5.4.       Ref. [23] seems incorrect.

Minor comments:

6.        P. 1, the first paragraph of the introduction. The phrase “rotation orientation and angle” looks unclear.

7.        P. 1, the paragraph “For example…”; p. 2, the paragraph “Wang et al…”. I recommend the authors provide a relevant reference in the first sentence of the paragraphs and not in the last.

8.        P. 2. When referring to other works, the authors use different styles, for example: “Jae-Hyeok Kim et al.,” “Wang et al.,” and “S. Yang et al.” I recommend the authors unify their citations. (Please check p. 1 too.)

9.        P. 2, the last paragraph. It is unclear what the authors mean under “the groove parameters are set in a pseudo-random manner.”

10.    P. 3, Fig. 1a:

10.1.   Rotations α, β, and γ are shown with a clockwise direction around the corresponding axes. Do the authors consider this direction? The common practice is to use a counter-clockwise direction. (Please check Figs. 1b and 2b too.)

10.2.   Shouldn’t the rod be shown along the rotated Z’ axis like in Fig. 2b?

11.    P. 3, the paragraph under Fig. 1. The authors mention the needs of the precision engineering field but do not specify any particular precision values.

12.    P. 3, the same paragraph. There should be “two methods” instead of “two method.”

13.    P. 3, Fig. 2a; p. 4, Fig. 3. The figures have label “Permanent magnets.” According to the text, there is only one magnet.

14.    P. 4, the paragraph above Fig. 3. The first, second, and fifth sentences are in an imperative mood, which looks inappropriate in a scientific article.

15.    P. 4, Fig. 3. Numbers (0, 4), (4, 0), and so on have no explanations.

16.    P. 4, the paragraph under Fig. 3. The first sentence looks incomplete.

17.    P. 4, the same paragraph. The authors introduce acronym RBF but do not spell it out explicitly.

18.    P. 4, the same paragraph. The authors mention the previously described spherical coding scheme, but there was no description of such a scheme.

19.    P. 4, the same paragraph. The term “positive significance” looks unclear.

20.    P. 4–5, Fig. 4. I recommend the authors improve the figure quality. At the moment, the figure becomes blurry when zooming.

21.    P. 6, Fig. 6. Similar to the comment above. In addition, the colorbars have no units, and the numbers are too small and difficult to read.

22.    P. 7, the paragraph above Fig. 7. The sentence “Based on the design…” looks incomplete.

23.    P. 10, subsec. 4.1. The authors introduce acronyms RPI, LS, and PI, but do not explain their meaning.

24.    P. 11, the first paragraph of subsec. 4.2. The phrase “results between regions are not significantly different” looks vague. I recommend the authors provide numerical values for this “insignificant difference” (for example, in percent).

25.    P. 11–12, Fig. 12. The colorbars have no units.

26.    P. 13, Tables 3 and 4. What are υR and υD? The text does not explain these parameters.

27.    P. 13, the line above Eq. (7). The term “synthesis freedom” is unclear.

28.    P. 14, the sentence under Table 5:

28.1.   The sentence is in an imperative mood, which looks inappropriate in a scientific article.

28.2.   What are k and t?

29.    P. 14, the author contributions section. Who is Y.Y.?

Comments on the Quality of English Language

There are several unclear sentences and terms that should be revised.

Reviewer 3 Report

Comments and Suggestions for Authors

(1) Due to existence similar studies that provide investigation of rotational capabilities of spherical joints, the authors are invited to highlight research novelty and originality of the proposed approaches.

(2) According to the schematic diagram of measurement scheme presented in Figure 2, the described method of measuring the rotational angle limits the movement of the hinge. This needs to be written in the text.

(3) Figures:

§  Figure 1: it is unclear why the X/Y/Z/ coordinate system has different colours in (a) and (b). The quality of this figure also might be increased. By the way the axes of the X/Y/Z/ coordinate system are inclined, however the ball is in initial position (normally axes of the XYZ and X/Y/Z/ should coincide);

§  Figure 4 has lower quality and it seems difficult to comprehend. Also Figures 6, 7b have weak quality;

§  Figures 6, 12: dimension of a physical quantity is absent;

§  Figure 12: the authors write that test points during calibration are also plotted. It seems they might be removed.

(4) In Section 4 the authors present graphs with angle errors. In this section brief conclusions based on the presented graphs should be presented.

(5) It is recommended to add original sources for equations (4)-(7).

(6) It looks like introduction includes some of the authors contribution. Normally introduction should include a review part. By the way, some phrases in the introduction look inappropriate, such as: “from Minnesota State University, USA”, “Korean scholars”, “from the Chinese Academy of Sciences”, etc. – it seems it is not necessary to mention this information as it is research paper.

(7) The authors are invited to polish text and English language. There are some misprints (for example, [1] and [2] in references duplicate each other). Also, it seems references [1-3] do not describe research idea.

Comments on the Quality of English Language

Editing of English language is recommended

Round 2

Reviewer 2 Report

Comments and Suggestions for Authors

The authors have considered all the issues and improved the paper quality. I have just a few minor comments to be addressed:

1.        P. 1, the first paragraph of the introduction. In the added sentence, I think the authors missed the word “joint” after the word “spherical.”

2.        P. 3–4, Figs. 1b and 2a. The figures have labels “Permanent magnets,” while there is only one permanent magnet.

3.        P. 4, Fig. 3. The authors write “The coordinates in Figure 3 represent the position of the sensor…” The authors, however, do not specify the units of the shown numbers. Are these coordinates in mm?

4.        P. 17, Ref. [23]. The authors have responded that this reference is in Chinese. I have found the paper titled “Optimized design and accuracy improvement of intelligent ball hinge” written by the same authors and published in the same issue with the same page numbering in Chinese Journal of Scientific Instrument. Isn’t it the same paper? If so, I recommend the authors update the information.

Reviewer 3 Report

Comments and Suggestions for Authors

The authors have answered all our comments. After the polishing a manuscript it might be accepted.

Comments on the Quality of English Language

The authors have answered all our comments. After the polishing a manuscript it might be accepted.

Author Response

Thank you very much for your comments on the manuscript. We have made appropriate modifications and refinements to the article.